# Stress and resilience of nursing students in clinical training during political violence: A palestinian perspective

Ibrahim Aqtam[1]*, Ahmad Ayed[2], Othman A. Alfuqaha[3], Mustafa Shouli[1]

1 Department of Nursing, Ibn Sina College for Health Professions, Nablus University for Vocational and Technical Education, Nablus, Palestine, 2 Faculty of Nursing, Arab American University, Jenin, Palestine, 3 Department of Counseling and Mental Health, The World Islamic Sciences and Education University, Amman, Jordan

* ibrahim.aqtam@nu-vte.edu.ps, info@nu-vte.edu.ps

## Abstract

### Introduction

Nursing is one of the most demanding undergraduate degrees because students must master rigorous theory while simultaneously developing hands-on clinical skills. In conflict zones, political violence magnifies these challenges during clinical training.

### Objective

To quantify perceived stress and resilience among Palestinian nursing students during political violence and examine their relationship and demographic associations.

### Methods

In this cross-sectional study, 310 students completed validated Arabic versions of the 29-item Perceived Stress Scale (PSS) and 10-item Connor-Davidson Resilience Scale (CD-RISC-10). Completion time averaged 10–15 minutes total. Analyses included Pearson correlation, $t$-tests, ANOVA, and multiple regression.

### Results

Mean stress was high ($81.1 \pm 7.3$) based on established PSS cutoffs, while resilience was moderate ($28.0 \pm 1.8$) according to CD-RISC-10 norms. A moderate positive correlation linked stress and resilience ($r = 0.314$, 95% CI [0.182, 0.442], $p < 0.001$). Male students reported higher stress than females ($t(308) = 2.20$, $p = 0.030$, Cohen's $d = 0.25$).

### Conclusion

Findings underscore the significant stress burden of political violence on nursing students and signal the need for targeted interventions that address both stress levels

**Data availability statement:** The datasets generated and/or analyzed during the current study are not publicly available due to participant confidentiality and privacy restrictions imposed by the Institutional Review Board (IRB) of the Arab American University, Palestine. However, de-identified data may be made available upon reasonable request. Requests should be directed to the Ethics Committee of the Faculty of Allied Medical Sciences at Arab American University (Email: irb@aaup.edu), which will review each request for compliance with ethical guidelines and data protection regulations.

**Funding:** The author(s) received no specific funding for this work.

**Competing interests:** The authors have declared that no competing interests exist

and the complex bidirectional relationship with resilience, tailored to demographic risk factors.

## Introduction

Nursing is one of the most demanding undergraduate degrees, melding intensive coursework with high-stakes clinical practice [1]. Nursing students commonly face stressors such as heavy academic workloads, patient-care responsibilities, and time pressures [2–4]. In politically unstable regions, these intrinsic demands are compounded by external threats, violence, restricted mobility, and economic precarity, which magnify psychological strain during clinical placements [5,6].

Stress involves complex psychophysiological processes. When a person appraises a situation as threatening, an amygdala-initiated cascade activates the hypothalamic-pituitary-adrenal (HPA) axis, releasing corticotropin-releasing hormone and adrenocorticotropic hormone, followed by sustained cortisol secretion. Simultaneously, sympathetic adrenal-medullary activation elevates catecholamines, raising heart-rate variability and blood pressure. Chronic activation of these pathways impairs hippocampal memory processing, immune function, and cognitive performance [7,8]. Such physiological dysregulation can be especially detrimental for health-care trainees who must maintain cognitive acuity and emotional stability in high-risk environments.

Resilience, the capacity to adaptively navigate adversity, has been extensively studied in healthcare education contexts [9]. Research suggests resilience may interact with stress in complex ways. While commonly conceptualized as protective, literature has documented various relationship patterns, including negative associations in stable environments [10] and occasionally positive correlations in high-adversity contexts [11].

Previous investigations using validated Arabic measures have established psychometric properties appropriate for use with student populations in the Middle East [12,13]. These assessment tools provide standard metrics that allow comparison with international norms.

Sex differences in stress responses involve both physiological and sociocultural factors. Physiologically, differences in HPA axis reactivity, autonomic nervous system activation patterns, and gonadal hormone influences create distinct stress response signatures [14]. Cortisol reactivity typically differs between males and females, with females often showing blunted responses to acute stressors but potentially greater vulnerability to certain chronic stressors [15]. Sociocultural, sociocultural expectations may significantly modulate how stress is expressed and experienced [16]. While global data often show higher stress in females, patriarchal norms in Palestine may heighten academic and financial expectations for males, altering stress expression [17,18].

Guided by the Transactional Model of Stress and Coping [19], this study aims to (1) measure stress levels and resilience among Palestinian nursing students, (2) examine the relationship between stress and resilience, and (3) identify demographic factors

associated with these psychological constructs. We hypothesize, based on previous literature in conflict settings [5,6] that Palestinian nursing students will (a) report stress levels exceeding global norms and (b) display moderate resilience levels (CD-RISC-10 scores of 25–30) consistent with previous studies of populations experiencing chronic adversity [20,21]].

## Methods

### Study design, inclusion, and exclusion criteria

A descriptive cross-sectional design measured perceived stress and resilience among nursing students during a period of political violence in Palestine.

Inclusion criteria consisted of enrolment in the Arab American University School of Nursing (second–fourth year); completion of ≥ 1 clinical-training course; and provision of informed consent. Exclusion criteria were: first-year students (no clinical exposure); students on academic leave or suspension; and individuals' self-reporting a psychiatric diagnosis.

### Setting and data collection

Data was collected on campus between 13 January 2025 and 13 February 2025. Of 336 students approached, 310 participated (response rate = 92.2%). Participants completed validated Arabic-language questionnaires during scheduled clinical-training sessions (taking approximately 10–15 minutes). No monetary incentive was offered; contributors were acknowledged.

### Study instruments

The study utilized the following instruments: The Perceived Stress Scale (PSS), a validated Arabic 29-item version (Sheu et al., 1997) using a Likert scale (0–4) for a total score range of 0–116, with scores > 75 indicating 'high' stress levels [22]; the Connor-Davidson Resilience Scale (CD-RISC-10), a validated Arabic 10-item version (Campbell-Sills & Stein, 2007) using a Likert scale (0–4) for a total score range of 0–40 [23], with scores between 25–30 representing '-moderate'- resilience in populations exposed to adversity [20]; and a demographics questionnaire capturing sex, age, and academic year.

### Pilot study

Content validity and reliability were established for the Arabic versions of both instruments. The content validity index was 0.94, with Cronbach's α = 0.89 for PSS and α = 0.92 for CD-RISC-10, confirmed in a 30-student pilot study. These values align with published Arabic validations of these instruments.

### Ethical considerations

Approval for this study was obtained from the Institutional Review Board (IRB) of Arab American University, Palestine (Approval Code: R-2025/A/3/N). The research adhered to the ethical principles outlined in the Declaration of Helsinki, ensuring respect for participant autonomy, beneficence, and justice. Participants provided written informed consent after being fully informed of the study's purpose, procedures, potential risks, and benefits. Confidentiality and anonymity were rigorously maintained: all data were stored in password-protected files accessible only to the research team, and identifiers were removed prior to analysis. Participants were assured of their right to withdraw at any time without penalty. No monetary compensation was provided, but students received acknowledgment for their contributions. Raw data are not publicly available to protect participant privacy, but de-identified datasets can be accessed upon reasonable request from the corresponding author.

### Statistical analysis

Data were analyzed using SPSS version 28. First, descriptive statistics (means ± standard deviation, frequencies, percentages) were calculated for all variables. Next, relationships between continuous variables (stress, resilience, and age) were examined using Pearson's correlation coefficient ($r$). Group differences were assessed using independent samples $t$-tests

(for comparing stress and resilience between sexes) and one-way analysis of variance (ANOVA) (for comparing across academic years).

For multivariate analysis, two separate multiple linear regression models were constructed to identify predictors of: (1) total stress and (2) total resilience. In each model, predictor variables included sex (coded as male = 1, female = 0), age (continuous variable, in years), and academic year (dummy coded with second year as reference category). Results are reported as adjusted $R^2$, standardized beta coefficients (β), and 95% confidence intervals.

Effect sizes were calculated as Cohen's *d* for *t*-tests and partial eta-squared ($\eta^2$) for ANOVA. Statistical significance was set at α = 0.05. All statistical parameters are reported in italics as per convention.

## Results

### Sample characteristics

A total of 310 nursing students completed the survey (response rate = 92.2%). Mean age was 21.4 ± 1.4 years (range 19–25); 54.8% were female and 45.2% male (Table 1).

### Perceived stress and resilience levels

Mean total stress was 81.1 ± 7.3, exceeding the PSS clinical threshold of 75 [20]. Resilience averaged 28.0 ± 1.8, falling within the "moderate" CD-RISC-10 range (25–30) identified in previous research [22]. Analysis of stress subscales showed that assignments/workload (mean score 3.2 ± 0.4) and environmental stress (mean score 2.8 ± 0.5) were the largest contributors to overall stress (Fig 1).

### Correlation between stress and resilience

A moderate positive correlation linked to total stress and resilience (*r* = 0.314, 95% CI [0.182, 0.442], *p* < 0.001; Fig 2). Among stress subscales, stress from patient-care demands correlated most strongly with resilience (*r* = 0.239, *p* = 0.001), followed by peer-interaction stress (*r* = 0.146, *p* = 0.010).

### Demographic differences

Male students reported significantly higher stress than females (82.1 ± 7.1 vs 80.3 ± 7.4; *t*(308) = 2.20, *p* = 0.030, Cohen's *d* = 0.25). Resilience did not differ by sex (*t*(308) = 1.10, *p* = 0.277). Age showed a weak positive association with stress (*r* = 0.132, *p* = 0.020). One-way ANOVA indicated no effect of academic year on stress (*F*(2, 307) = 0.27, *p* = 0.762, $\eta^2$ = 0.002) or resilience (*F*(2, 307) = 2.04, *p* = 0.132, $\eta^2$ = 0.013) (Table 2).

### Regression analysis

The multiple regression model predicting total stress explained 3.5% of variance (adjusted $R^2$ = 0.035). Sex (β = 0.192, 95% CI [0.023, 0.361], *p* = 0.026) and age (β = 0.145, 95% CI [0.008, 0.282], *p* = 0.038) significantly predicted higher stress;

**Table 1. Demographic Characteristics (N = 310).**

| Characteristic | Value |
| --- | --- |
| Sex | Male: 140 (45.2%), Female: 170 (54.8%) |
| Academic Year | Second: 101 (32.6%), Third: 105 (33.9%), Fourth: 104 (33.5%) |
| Age | Mean ± SD: 21.4 ± 1.4 years |
| Age Range | 19–25 years |

**Note:** SD = standard deviation.

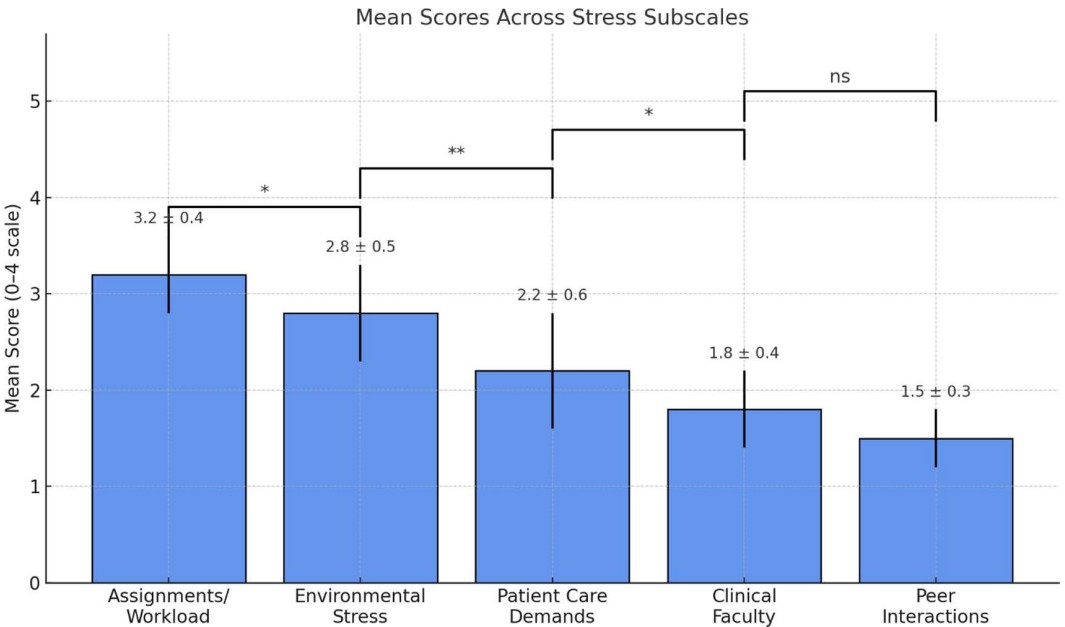

**Fig 1. Bar chart illustrating stress subscale contributions, highlighting assignments/workload (mean score 3.2±0.4) and environmental stress (mean score 2.8±0.5) as primary stressors.**

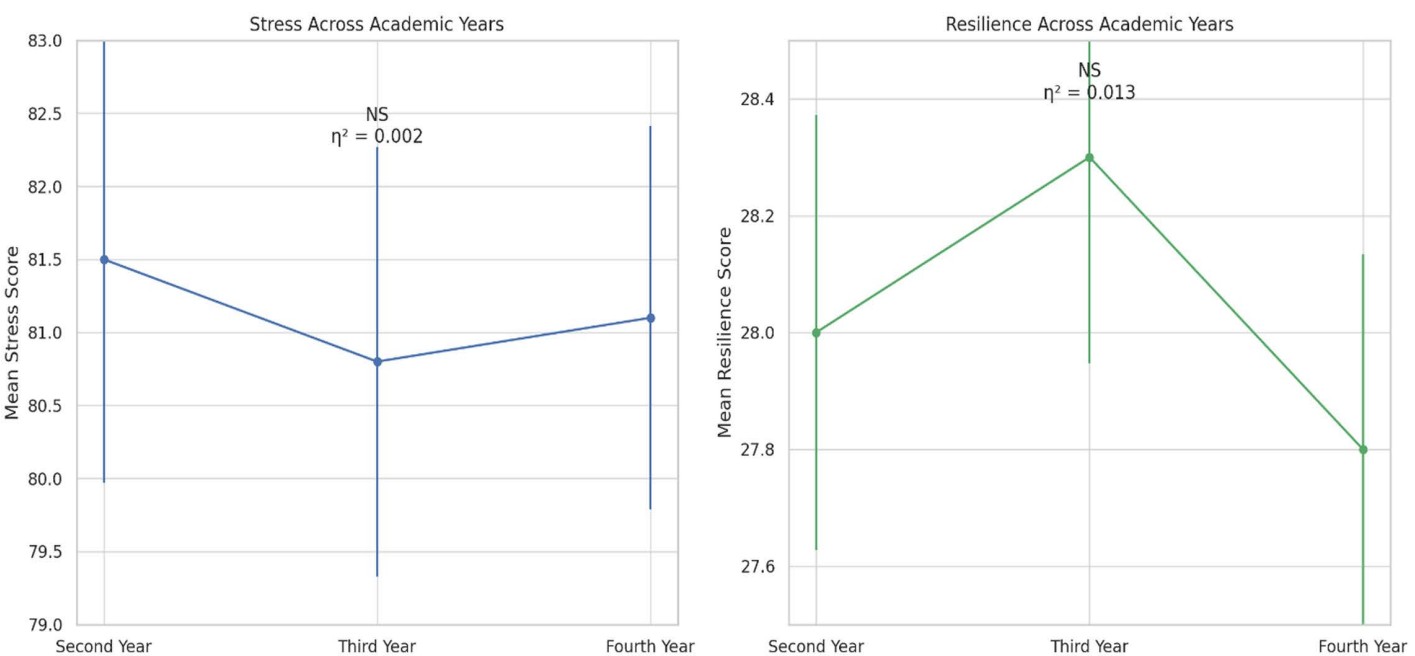

**Fig 2. Scatter plot demonstrating the moderate positive correlation between total stress and resilience scores (*r*=0.314, *p*<0.001).**

**Table 2. Demographic Differences in Stress and Resilience.**

| Variable | Stress (Mean ± SD) | Resilience (Mean ± SD) | Test Statistic | p-value |
|---|---|---|---|---|
| **Male** | 82.1 ± 7.1 | 28.1 ± 1.9 | $t = 2.20$ | 0.030 |
| **Female** | 80.3 ± 7.4 | 27.9 ± 1.8 | | |
| **Academic Year** | | | $F = 0.272$ | 0.762 |

**Note:** $\eta^2$ = partial eta-squared; Cohen's $d$ = effect size for $t$-tests.

academic year was non-significant ($\beta = -0.076$, $p = 0.281$). The model predicting resilience was non-significant (adjusted $R^2 = 0.009$); none of the predictors reached $p < 0.05$ (Table 3).

## Discussion

This study examined perceived stress and resilience among Palestinian nursing students undergoing clinical training amid political violence. Key findings include elevated stress levels classified as 'high' (81.1 ± 7.3) relative to established norms [22], moderate resilience scores (28.0 ± 1.8), and a counterintuitive positive correlation between stress and resilience ($r = 0.314$, $p < 0.001$). Below, we contextualize these results, address their implications, and acknowledge limitations.

### Stress in conflict zones

The observed stress levels align with prior research in conflict-affected regions [6,24], where political instability exacerbates academic and clinical pressures. Environmental stressors, checkpoints, safety concerns, and restricted mobility, contributed significantly to stress scores (2.8 ± 0.5), underscoring how geopolitical realities amplify training challenges. Notably, assignments/workload emerged as the highest stress subscale (3.2 ± 0.4), consistent with global studies highlighting academic demands as primary stressors for nursing students [4,25]. Persistent elevation of stress hormones and sympathetic nervous system activity in conflict zones may impair cognitive function, emotional regulation, and immune responses [7,26], with particular relevance to contexts of chronic political violence where HPA axis dysregulation can become prolonged and maladaptive [8], warranting urgent institutional interventions.

### Resilience and its complex relationship with stress

Resilience levels (28.0 ± 1.8) fell within the moderate range observed in populations exposed to chronic adversity [20]. Contrary to conventional assumptions that resilience functions primarily as a protective buffer against stress, our findings revealed a positive correlation between these constructs. This seemingly paradoxical relationship challenges simplistic conceptualizations of resilience as merely stress-reducing and underscores its context-dependent nature. Specifically, the bidirectional interaction suggests that resilient individuals may engage more proactively in challenging environments, assuming greater responsibilities (e.g., volunteering for high-risk rotations or leadership roles during crises) that inadvertently increase stress exposure [21]. Such proactive engagement aligns with prior literature documenting shifting stress-resilience dynamics, where negative associations in stable environments transition to positive correlations in

**Table 3. Regression Models for Stress and Resilience.**

| Predictor | Stress (β) | p-value | Resilience (β) | p-value |
|---|---|---|---|---|
| Sex (Male) | 0.192 | 0.026 | 0.062 | 0.268 |
| Age | 0.145 | 0.038 | −0.018 | 0.733 |
| Academic Year | −0.076 | 0.281 | 0.078 | 0.209 |

**Note:** β = standardized beta coefficient; adjusted $R^2$ = variance explained.

high-adversity contexts where stressors become unavoidable [11]. These findings emphasize that resilience in conflict settings may reflect adaptive engagement with stressors rather than passive mitigation, necessitating interventions that acknowledge this complexity.

### Sex differences in stress response

Male students reported higher stress levels than females ($t$(308) = 2.20, $p = 0.030$), diverging from global trends where females typically report greater stress [17]. In Palestinian society, sociocultural expectations often place disproportionate academic and financial responsibilities on males, potentially amplifying stress despite lower emotional disclosure norms [17]. Sex-based physiological differences in stress response also merit consideration. Research has documented different patterns of HPA axis activation, with males typically showing stronger initial cortisol responses but potentially different habituation patterns to chronic stressors [15]. Autonomic nervous system reactivity also differs, with males often demonstrating stronger sympathetic responses [14]. These biological factors interact with sociocultural expectations to shape sex-specific stress profiles.

### Implications for nursing education

The findings underscore the complexity of stress-resilience dynamics in conflict settings and suggest several important implications for nursing education. For instance, structured mentorship programs could help students navigate clinical responsibilities without overwhelming engagement, while simulation-based training, such as virtual scenarios replicating conflict-zone challenges, may reduce environmental stressors identified in the study. Sex-specific support systems are critical to addressing unique pressures on male students in patriarchal settings, where sociocultural norms exacerbate stress despite global trends favoring females. Policymakers must prioritize funding for mental health services and safe clinical environments in conflict zones, ensuring nursing students train without compounding existential risks. Collaboration between universities and healthcare institutions could foster trauma-informed curricula and crisis response protocols, aligning education with the realities of political violence.

### Limitations and future directions

This study has several limitations: the cross-sectional design precludes causal inferences about stress-resilience dynamics, self-report bias risks over- or underestimation of psychological states, and single-institution sampling limits generalizability to broader conflict-affected populations. To address these constraints, future research should employ longitudinal designs to track stress trajectories and resilience outcomes over time in conflict zones. Mixed-methods approaches could elucidate cultural narratives and physiological mediators (e.g., cortisol levels, heart rate variability) of stress responses, while multi-institutional samples would enhance external validity and capture regional variability in political violence impacts.

## Conclusion

Political violence imposes a unique stress burden on nursing students, necessitating interventions that address both stress and resilience without oversimplifying their complex relationship. Our findings challenge the notion of a simple inverse relationship between resilience and stress, instead revealing a more nuanced bidirectional interaction where resilient individuals may experience higher stress levels in conflict settings. By integrating demographic-specific strategies, fostering supportive environments, and advocating for policy reforms, educators can better prepare students to thrive in high-risk clinical settings. Resilience development remains important but should be pursued alongside systemic efforts to reduce structural stressors in conflict-affected educational environments.

## Supporting information

**S1 Supplementary Material. Arabic versions of the Perceived Stress Scale (PSS) and Connor-Davidson Resilience Scale (CD-RISC-10).** Scoring guidelines with normative thresholds for classification.
(DOCX)

**S2 Supplementary Material. Detailed statistical analyses include correlation matrices, regression diagnostics, and effect size calculations.**
(DOCX)

## Acknowledgements

The authors would like to express their thanks to the students who participated in the study.

## Author contributions

**Conceptualization:** Ibrahim Aqtam, Ahmad Ayed, Othman A. Alfuqaha, Mustafa Shouli.

**Data curation:** Ibrahim Aqtam, Ahmad Ayed.

**Formal analysis:** Ibrahim Aqtam, Othman A. Alfuqaha, Mustafa Shouli.

**Investigation:** Ahmad Ayed, Othman A. Alfuqaha, Mustafa Shouli.

**Methodology:** Ibrahim Aqtam, Ahmad Ayed, Othman A. Alfuqaha.

**Project administration:** Ibrahim Aqtam.

**Supervision:** Mustafa Shouli.

**Writing – original draft:** Ibrahim Aqtam, Ahmad Ayed.

**Writing – review & editing:** Ibrahim Aqtam, Othman A. Alfuqaha, Mustafa Shouli.

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
