## [Decision Letter · Decision Letter 0]

May 09 2025

Dear Dr. aqtam,

Thank you for submitting your manuscript to PLOS ONE. After careful consideration, we feel that it has merit but does not fully meet PLOS ONE’s publication criteria as it currently stands. Therefore, we invite you to submit a revised version of the manuscript that addresses the points raised during the review process.

We look forward to receiving your revised manuscript.

Kind regards,

Mohamed Gamal Elsehrawy

Academic Editor

PLOS ONE

Reviewers' comments:

Reviewer's Responses to Questions

**Comments to the Author**

1. Is the manuscript technically sound, and do the data support the conclusions?

Reviewer #1: Yes

Reviewer #2: Yes

Reviewer #3: No

Reviewer #4: Yes

2. Has the statistical analysis been performed appropriately and rigorously?

Reviewer #1: No

Reviewer #2: Yes

Reviewer #3: Yes

Reviewer #4: Yes

3. Have the authors made all data underlying the findings in their manuscript fully available?

Reviewer #1: Yes

Reviewer #2: Yes

Reviewer #3: No

Reviewer #4: Yes

4. Is the manuscript presented in an intelligible fashion and written in standard English?

Reviewer #1: Yes

Reviewer #2: No

Reviewer #3: Yes

Reviewer #4: Yes

Reviewer #1: Review of the Manuscript ID PONE-D-25-08134 Title: “Stress and Resilience of Nursing Students in Clinical Training during Political Violence: Palestinian Perspective” for the Plos One Journal.

General Comments

From my point of view, it is a very interesting topic and simultaneously it seems that to the best of my knowledge is an empirical research aims to measure perceived stress and resilience among nursing students during periods of political violence in Palestine. Moreover, it aimed to investigate the association between nurse’s demographic characteristics and both stress and resilience. In this study, 310 nursing students were conveniently selected in Palestine between January 13, 2025, and February 13, 2025. We used two validated Arabic scales: Perceived Stress Scale and Connor-Davidson Resilience Scale. Pearson correlation, t-test, and ANOVA test were conducted. The findings show that the political violence psychological effect is so strong that it calls for nursing students' psychological interventions that are stress-reducing and build resilience. Nursing educators and institutions have the responsibility to use strategies that can help such challenges for better mental health and academic success during such politically turbulent times.

The paper consists of following sections: Introduction, Methods, Results, Discussion, Implications for Nursing Education, Limitations, Conclusion.

However, I find some recommendations:

1. The Manuscript needs careful English proofreading because there are some shortcomings. For instance, the article “the” is sometimes missing in front of nouns, the message in some paragraphs is not clear enough. It looks like the first part was written by one author with a greater command of the English language, and the rest of the paper was written by someone else. The numerous grammar errors made this a difficult paper to read. It was strange to see the authors refer to tables that were not submitted. I was unable to find any supplementary material to the submission, so I think this was truly omitted by the authors. Please read the manuscript carefully.

2. It would be very useful to add in the "Introduction" section the purpose, objectives and hypothesis of the research. I consider that a weak point of the paper is that the authors did not show the novelty of the paper compared to other works. That is why, I consider that the introduction should specify the novelty of the paper compared to other papers published in this area.

3. The authors must also show the values of the adjusted R-square, which is more relevant in the methods used in this paper.

4. Authors must present the results of the processing in tabular form and interpret the results. The paper cannot be accepted in this form.

5. The conclusions at the end of the paper should be expanded showing the policy implications of the research results.

In conclusion, the article should be improve. It should also be enhanced with a review of the literature adequate to the subject and a broader interpretation and commentary of the research results.

Reviewer #2: It recommended to revise and re-edit whole the manuscript by English professional in Medical sciences.

Some sentences have grammatical errors( Three hundred and ten out of 310 nursing students completed the paper-based survey successfully,

resulting in a response rate of 92.2%

Reviewer #3: This manuscript examines self-reported stress and resilience in nursing students (2nd to 4th academic year) in an environment of political violence/war in Palestine. The authors measured levels of these variables with the Perceived Stress Questionnaire (PSS) and the Connor-Davidson Resilience Scale (CD-RISC-10), respectively. Stress and resilience were then related to each other, and the authors further investigated effects of sex and academic year (among other demographic variables) on these variables. The results showed that stress levels were higher up on the scale, whereas resilience levels were describes as being moderate. Stress and resilience correlated positively with each other, and stress levels were further influenced by sex (higher in males) and age (higher with older age). The authors conclude that it is important to incorporate interventions aimed at increasing resilience and thereby reduce stress of nursing students.

Overall, the study examines an important topic in a difficult political situation, and I highly appreciate the effort, especially because it appears to be properly powered as reflected by the high sample size. I like the idea of examining stress and resilience levels in an environment of political violence. However, there are several concerns that dampen my enthusiasm for this manuscript.

Major concerns

Major concern 1: My first major concern is about the validity of the research questions. The authors describe that they aim to “measure perceived stress and resilience among nursing students during periods of political violence in Palestine” (p.6) and to “to investigate the association between nurse’s demographic characteristics and both stress and resilience” (p.6). I will refer to the former aim as RQ1 and to the latter as RQ2.

In my viewpoint, RQ1 is not a valid research question because it only deals with measuring levels of a variable. A research question that could be asked here is whether these levels are different compared to political stable times (alternative 1) or whether these levels are different compared to other subpopulations, e.g., students of mathematics, during periods of political violence (alternative 2). There are more research questions that one can think of and each of these needs a control or comparison group, which is not available here, and thus the mentioned alternatives cannot be investigated with the current data set. Therefore, I think that measuring stress and resilience levels and presenting only the descriptive data without a comparison group, is only valid if there are conventions or standardized values of these scales which provide values to compare against. The authors speak of “high levels” of stress and “moderate levels” of resilience, but it is not argued, why the measured levels are classified as “high” or “moderate”, respectively. To justify the classification into these categories, it is strongly advised to present validated comparison values. RQ2 on the other hand, is a valid research question, and its analysis was conducted appropriately. However, the analysis of the correlation between stress and resilience levels, which appears to be one of the main analyses, is not mentioned explicitly in the research questions.

Major concern 2: My second major and maybe biggest concern is that the drawn conclusions overreach and are not supported by the data. Specifically, the relationship between stress and resilience is reported to be positive, which is against the idea that resilience is a protective factor for stress. However, even though there is a short paragraph that discusses this surprising finding in a somewhat plausible way, claiming that higher resilience may lead to more responsibility and takeover of more tasks, which in turn can lead to more stress, all further paragraphs draw conclusions that cannot be supported by the finding of this study, but instead, would require the opposite finding of a negative correlation. For instance, in the discussion it is claimed that:

- “While resilience serves a protective function, it is not a panacea. Results of the current study and those from other studies point toward the need for institutional support, mental health services, and culturally adapted interventions [15,38]. It is very important in enabling nursing students insuch politically unstable regions to deal with specific challenges” (p.13) or that

- “The findings of this study suggest that nursing curricula should be aimed at strengthening the resilience of nursing students.” (p.13) or that

- “The findings highlight the importance of resilience as a protective factor against the development of stress and recommend targeted interventions to support students.” (p.14)

The observed positive correlation between stress and resilience levels does not support any of these conclusions. Instead, it would rather suggest the opposite, claiming that reducing resilience levels should be associated with lowered stress levels. Moreover, because this analysis is correlational, the drawn conclusions need to be taken with caution in any case, even if they were supported by the data.

Major concern 3: My third major concern is that, in the introduction, the balance is strongly biased towards resilience, whereas basics of stress are not provided. More specifically, a definition of stress, the activation of typical stress response systems, physiological markers, and the relationship between physiological markers and self-reports, would be a good starting point to introduce basic knowledge about stress research. In this context, the parts about resilience could be compressed, as they appeared to be repetitive in some respects.

Major concern 4: My fourth major concern is that the description and reporting of statistical analyses is not complete, and the interpretation is not always valid. As already mentioned, for RQ1, descriptive values are used to classify into “high” or “moderate” levels and this classification is used to speak of a psychological effect that “is so strong that it calls for nursing students' psychological interventions that are stress-reducing and build resilience.” (p.3). Here, it is claimed that a psychological effect is “strong”, even though effect sizes have not been calculated nor reported. Moreover, it is not clear what this effect refers to: as previously mentioned, the stress and resilience levels have not been compared to a control group or standardized values.

Further, in the method and in the result section, it should be added which variables are used for the t-test and anova. Moreover, statistical parameters are not provided completely, and they should always contain (at least).:

- For correlation analyses: r and p (sometimes, only p was reported)

- For t-tests: t, df, p, and a measure of effect size, e.g., cohen’s d (df and effect sizes were not reported)

- For anovas: F, df, p, and a measure of effect size, e.g., partial eta-squared or generalized eta-squared (df and effect sizes were not reported)

Additionally, in the results, data is always shown using tables. Even though this is appropriate for tables 1-3, for which I would advise more detailed captions (e.g., mentioning the questionnaires or the fact, that it describes self-reports of stress and resilience), I would advise to plot the results of table 4 to have a visualization of the results of the t-test and the anova. Lastly, the interpretation of results should be based on the data, which is not the case (as described in major concern 2, but also in some other instances like the above mentioned “strong” psychological effect).

Major concern 5: According to the data availability statement, data will not be made available publicly (or: are claimed to be found in the manuscript and its supporting information files). For the raw data, this is not the case, so other researchers cannot conduct the same analyses.

Minor concerns

Minor concern 1: I would advise to not use the term “psychological distress”. There is a lot of confusion about this term in the literature and according to Bienertova-Vasku et al. (2020), it does not add any information compared to just using “stress”.

Minor concern 2: In the introduction, it is said that “Resilience has thus been proposed as a modulator of the influence on the quality of life of students and as a mitigator of psychological distress, especially in periods of crisis [16,17].“ (p.5).

Here it is unclear, which effect resilience is modulating. There needs to be a modulator, an independent variable and a dependent variable, but in the sentence, there is only the potential modulator (resilience) and the dependent variable (quality of life).

Minor concern 3: The cited original article for development of the PSS and CD-RISC-10 should be corrected (p.7). For PSS, it is “Sheu, S., Lin, H., Hwang, S., Yu, P., Hu, W. and Lou, M. (1997) The Development and Testing of Perceived Stress Scale of Clinical Practice. Nursing Research (Republic of China), 5, 341-351.” And for CD-RISC-10, the reference number 19 should be corrected to 18. Because there is apparently some confusion with the numbers, it would be generally recommended to check whether references are correctly assigned.

Minor concern 4: A more detailed description of the testing procedure, including concrete steps, total session lengths, compensation etc. would be helpful

Minor concern 5: In the statistical analysis part (p.8), it is more common to refer to α-level instead of p-level

Minor concern 6: In the discussion (p.12), resilience levels are directly compared to two studies (Labrague et al. 2021 & Smith et al., 2017), which, however, used different scales. This comparison does not appear valid and it is also unclear, what exactly “the rating is somewhat lower” refers to. Does it mean that the rating from this study are lower or from the other studies? The next sentence reads as if the latter is true, and, in this case, the result itself appears to be surprising and would need more discussion

Minor concern 7: In the discussion (p.12), sex-differences are mentioned and compared to one Nigerian study. Here, more extensive elaboration on general sex differences in stress measures and the stress response is warranted.

Minor concern 8: The authors use the terms “gender-differences” and “male/female”. When the focus is on biological differences, it is advised to talk about sex-differences, which are commonly related to being male or female. When the focus is on identity, societal roles or cultural expectations, it is advised to talk about gender-differences; in this case, however, I would not speak of males/females, but of man/woman.

Minor concern 9: In the supplement, there are no questionnaires, even though it is referred to them in the manuscript. Also, it would be helpful to add the age range of participants.

Minor concern 10: The terms stress and stressors are confused, sometimes the authors talk about high levels of stress, and sometimes about high levels of stressors (as in the beginning of the discussion). The PSS clearly measures levels of stress

Minor concern 11: The authors present a sample size calculation using RaoSoft, which I do not know about, but it seems valid. However, it appears that the term “response rate” is used instead of “response distribution”, which could be corrected.

Minor concern 12: In the results (p.8), it is said that “Three hundred and ten out of 310 nursing students completed the paper-based survey successfully”. The second 310 should be changed to 326

Minor concern 13: In the discussion (p.14), one paragraph ends with “As such,”, This should be corrected.

Miscellaneous remarks

A few further points that could be considered to increase the quality of the manuscript:

- Which of the subscales from the PSS would be expected to be high in presence of political violence? Is the pattern of results reflecting this?

- The conclusion and discussion of sex-differences and relationship between stress and resilience is interesting. Here, I advise to add literature about sex differences in the physiological stress response, which likely also plays a role (even though the societal aspect seems plausible, it is probably not sufficient)

- T-tests and ANOVA could also be conducted for exploring whether there are differences in the subscales of the PSS (especially for sex differences, as the authors observe an overall effect here)

- The manuscript is generally not written badly, but in some parts, I had the feeling that language editing (and in some cases, editing of the formal report of statistics) could be helpful

Reviewer #4: The paper addresses an important topic related to stress and resilience. However, some areas need improvement. Below is some feedback for improvement.

- Add literature review for second objective association between nurse’s demographic characteristics and both stress and resilience (eg: gender, age)

- State what is the theory being used in this study.

- Methodology- What is the inclusion and exclusion criteria for the targeted sample.

- Add more demographic characteristics in the table related to the study (eg: age)

- Limitation, hanging sentence. “As such”

- Add implication and conclusion related to the second objective of the study

**Do you want your identity to be public for this peer review?** For information about this choice, including consent withdrawal, please see our Privacy Policy

Reviewer #1: No

Reviewer #2: No

Reviewer #3: No

Reviewer #4: No

---

## [Author Response · Author response to Decision Letter 1]

27 Mar 2025

Dear Editor’s and Reviewers,

Thank you for your valuable feedback on our manuscript. We have carefully reviewed and addressed all comments and suggestions. The necessary revisions have been made to enhance the clarity, accuracy, and overall quality of the manuscript.

Please find the revised version attached for your review. We appreciate your time and consideration and look forward to your feedback.

Best regards,

Dr Aqtam

---

## [Decision Letter · Decision Letter 1]

Jun 07 2025

Dear Author,

Thank you for submitting your manuscript to PLOS ONE. After careful consideration, we feel that it has merit but does not fully meet PLOS ONE’s publication criteria as it currently stands. Therefore, we invite you to submit a revised version of the manuscript that addresses the points raised during the review process.

We look forward to receiving your revised manuscript.

Kind regards,

Mohamed Gamal Elsehrawy

Academic Editor

PLOS ONE

Reviewers' comments:

Reviewer's Responses to Questions

**Comments to the Author**

Reviewer #1: All comments have been addressed

Reviewer #3: (No Response)

2. Is the manuscript technically sound, and do the data support the conclusions?

Reviewer #1: Yes

Reviewer #3: Yes

3. Has the statistical analysis been performed appropriately and rigorously?

Reviewer #1: Yes

Reviewer #3: Yes

4. Have the authors made all data underlying the findings in their manuscript fully available?

Reviewer #1: Yes

Reviewer #3: No

5. Is the manuscript presented in an intelligible fashion and written in standard English?

Reviewer #1: Yes

Reviewer #3: Yes

Reviewer #1: Accept for publication because the authors took into consideration all the recomandations made by reviwer.

Reviewer #3: The manuscript has improved in certain aspects, and several of my previous concerns have been addressed. However, I believe that further changes are necessary to meet methodological standards. The current revision gives the impression of being somewhat rushed, and some of the concerns were not thoroughly elaborated on. I recommend that the authors provide more detailed and precise responses to the remaining issues and revise the manuscript accordingly. Lastly, in a few cases, the responses to concerns did not appear to align with the (not existent) changes in the manuscript. Before evaluating the responses to my previous concerns, I will state 3 new (minor) concerns.

New concern 1

Generally, the abstract seems to be unchanged (apart from the addition of statistical parameters, which should be italic)

• same conclusion (“strong psychological effect”) as before? The assumption of a “strong effect” is not based on data (effect sizes)

• In the abstract, it is said that validated Arabic scales were used. In the methods, it is emphasized, that the questionnaire was in English and took 10-15 minutes. It is confusing and unclear, if an Arabic or English questionnaire was used. Also, the sentence suggests that only one questionnaire was completed, even though it should be at least two

New concern 2

In the introduction, it now says: “The study hypothesizes that nursing students experiencing political violence exhibit higher levels of stress and moderate resilience.” This hypothesis appears to coincidentally align with the found pattern and seems to be articulated post-hoc (as it was not explicitly stated in the initial submission and is also not backed by research, at least the “moderate resilience” part).

New concern 3

Redundancy problems:

• Remove redundancy: This sentence appears twice: “See Supplementary Material S1 for the full questionnaire details, including the Perceived Stress Scale (PSS) and the Connor-Davidson Resilience Scale (CD-RISC-10).)“

• In the discussion, there is now a new paragraph discussing the surprising positive correlation between stress and resilience (starting with “contrary to conventional assumptions …”). However, the following paragraph (which is older) basically repeats the information. Please remove the redundancy.

Major concerns

Major concern 1: The justification of classifications by normative values is an improvement and a step into the right direction. However, I think that this should be incorporated more explicitly in the manuscript. Especially because the cited paper for the Resilience Scale, which indeed provided values for different population groups, used the original version with 25 items. Thus, it needs to be elaborated on which values for the shorter scale may relate to the ones presented in the original publication and thereby justify the classification into “moderate”. Also check out this paper: https://doi.org/10.1016/j.jpsychires.2009.01.013. Besides, the term “moderate” was not used in the Connor & Davidson (2003) paper. Moreover, these methodological details about classification in “high” or “moderate” levels should not be mentioned in the introduction (as is done currently), but rather in the methods or results section (and discussed in the discussion section).

Furthermore, the analysis of the correlation between stress and resilience levels, which appears to be one of the main analyses, is not mentioned explicitly in the research questions (nor in the methods section).

Major concern 2: I acknowledge the properly changed main conclusion, however, there are still instances that imply the non-supported previous conclusion, e.g. in the conclusion-section: “The findings highlight the importance of resilience as a protective factor against stress.” Also, the implications for nursing curricular or nursing education still transfer the previous interpretation that is not supported by the data. I highly recommend not to imply that the data supports the role of resilience as a protective factor against stress (this should be consistently done everywhere in the manuscript).

Major concern 3: There is now one sentence about the physiological background of stress in the introduction. In my view, this should be extended according to the feedback that I gave after initial submission.

Major concern 4: The description and reporting of statistical analyses improved and now includes effects sizes and degrees of freedom. Also, two figures were incorporated to better visualize the results of the t-test and the ANOVA. It can still be optimized though.

• Statistical parameters in Latin letters should be formatted as italic.

• Boxplots are not the best way to visualize t-test or ANOVA results, as these analyses use the mean value and the variance, which is not clearly depicted in boxplots.

• The statistical analysis part of methods still lacks a precise and structured description of the conducted analyses, and, surprisingly, already includes some of the results, which belong to the results section

• The column “frequency” in table 1 cannot be used for mean, sd, or range of the variable age

• Figure 2 is placed above figure one and not next to its legend

• In the results, the regression analysis is not sufficiently described. How did the regression models exactly looked like? From the statistical parameters, one can guess that the predictors sex, age and academic year were incorporated as predictors in one model (respectively, for stress and resilience), but this information needs to be explicated. This analysis was also not mentioned in the methods section

I strongly advise providing a better structure of the statistical analysis part of the methods, the analyses conducted and the results.

Major concern 5: The data availability statement was changed appropriately.

Minor concerns

Minor concern 1: The term psychological distress is still widely used. The response of the authors implies that this is done only in certain cases, but I do not see any consistency here.

Minor concern 2: changed appropriately.

Minor concern 3: The citation for the CD-Risk-10 refers to the original CD-Risk with 25 items. I strongly advise citing the correct literature. In this case: 10.1002/jts.20271

Minor concern 4: changed appropriately.

Minor concern 5: changed appropriately.

Minor concern 6: changed appropriately.

Minor concern 7: More extensive elaboration on general sex differences in stress measures and the stress response was provided by adding a sentence with no sources and mentioning the main biological aspect (“hormonal differences”). I advise to be more detailed here.

Minor concern 8: It is still talked about “gender”-differences, although the focus seems to be on differences between males and females, i.e., sex-differences. The response to my concern suggests that this was changed, but it remains unchanged in the manuscript.

Minor concern 9: The supplement now includes the questionnaires used, but the supplementary material in general is confusing. It has now three different files and it appears that tables s1, s2 etc. refer to the tables in the manuscript and repeat the information. If it does not provide any additional content, it appears redundant.

Minor concern 10: The terms stress and stressors are now better differentiated, but check these instances:

• “there may be stresses”? (p.6)

• “Male nursing students are prone to have more stressors compared to female nursing students.“ -> stressors or stress? (p. 15)

Minor concern 11: The advised change from “response rate” to “response distribution” was not done appropriately (at a wrong instance). “50% response rate” in the sample size calculation should be adapted to response distribution. However, the 92.2% response rate of students completing the questionnaire was correctly named before and should not be changed to response distribution. (p.7-8)

Minor concern 12: changed appropriately.

Minor concern 13: changed appropriately.

Miscellaneous remarks

The language of the manuscript has improved, but there is still room for improvement and some obvious errors should be corrected (e.g., first sentence in abstract).

**Do you want your identity to be public for this peer review?** For information about this choice, including consent withdrawal, please see our Privacy Policy

Reviewer #1: No

Reviewer #3: No

---

## [Author Response · Author response to Decision Letter 2]

24 Apr 2025

Dear Editor’s ,

Thank you for forwarding the reviewers’ Stress and Resilience of Nursing Students in Clinical Training during Political Violence: A Palestinian Perspective.”

We have carefully considered each point raised and have provided detailed, point‑by‑point responses in the accompanying “Response to Reviewers” document. All suggested revisions have been incorporated into the manuscript, and corresponding changes are clearly marked.

Please let us know if any additional information is required.

Kind regards,

Dr Aqtam, on behalf of all co‑authors

---

## [Decision Letter · Decision Letter 2]

Jun 17 2025

Dear Dr. Aqtam,

Thank you for submitting your manuscript to PLOS ONE. After careful consideration, we feel that it has merit but does not fully meet PLOS ONE’s publication criteria as it currently stands. Therefore, we invite you to submit a revised version of the manuscript that addresses the points raised during the review process.

We look forward to receiving your revised manuscript.

Kind regards,

Mohamed Gamal Elsehrawy

Academic Editor

PLOS ONE

Journal Requirements:

Reviewers' comments:

Reviewer's Responses to Questions

**Comments to the Author**

Reviewer #1: All comments have been addressed

Reviewer #3: (No Response)

2. Is the manuscript technically sound, and do the data support the conclusions?

Reviewer #1: Yes

Reviewer #3: Yes

3. Has the statistical analysis been performed appropriately and rigorously?

Reviewer #1: Yes

Reviewer #3: Yes

4. Have the authors made all data underlying the findings in their manuscript fully available?

Reviewer #1: Yes

Reviewer #3: No

5. Is the manuscript presented in an intelligible fashion and written in standard English?

Reviewer #1: Yes

Reviewer #3: Yes

Reviewer #1: All comments have been successfully addressed, and the authors have provided comprehensive responses that strengthen the rigor of their analysis. Key methodological considerations have been effectively incorporated and clarified. Furthermore, robustness checks and supplementary analyses have been conducted where necessary, reinforcing the validity of the study's findings.

Given these improvements, the manuscript is well-prepared for publication. The authors have demonstrated a thorough understanding of the relevant literature and policy implications, ensuring that the study makes a valuable contribution to the field. The revisions provide clarity and depth, enhancing the overall quality of the work.

Reviewer #3: The manuscript has strongly improved and my concerns have been addressed. The supplementary material also was reorganized and now provides additional, non-redundant information. I only have a few minor remarks.

• The abbreviation for HPA is introduced twice in the introduction, for the second time being mentioned, it is sufficient to just use HPA

• bullet points in methods should be avoided, except it is accepted or desired by PlosOne

• Significance indicator in fig. 1 should more clearly show that the difference between bars is significant (it appears as if only the left bar is significantly different from zero)

• Also, the y-axes for fig.1 are too narrow and thus overemphasize differences. They should be adjusted accordingly

**Do you want your identity to be public for this peer review?** For information about this choice, including consent withdrawal, please see our Privacy Policy

Reviewer #1: No

Reviewer #3: No

---

## [Author Response · Author response to Decision Letter 3]

3 May 2025

Dear Editor and Reviewers,

We would like to express our sincere appreciation for your thoughtful and constructive feedback on our research.

We have carefully considered all the comments and suggestions provided in the reviewers' report. In response, we have revised the manuscript accordingly and have provided a detailed point-by-point response in the attached “Response to Reviewers” document. We believe that these revisions have significantly improved the clarity and overall quality of our manuscript.

We are grateful for the time and effort the reviewers and the editorial team have devoted to reviewing our work, and we appreciate the opportunity to revise and resubmit our paper.

We hope the revised manuscript meets your expectations, and we respectfully submit it for your further consideration.

Thank you once again for your valuable comments and support.

Sincerely,

Dr Aqtam

---

## [Editor Report · Decision Letter 3]

Stress and Resilience of Nursing Students in Clinical Training during Political Violence: A Palestinian Perspective

PONE-D-25-08134R3

Dear Dr. Ibrahim Aqtam,

We’re pleased to inform you that your manuscript has been judged scientifically suitable for publication and will be formally accepted for publication once it meets all outstanding technical requirements.

Kind regards,

Fadwa Alhalaiqa

Academic Editor

PLOS ONE
---

## [Editor Report · Acceptance letter]

PONE-D-25-08134R3

PLOS ONE

Dear Dr. Aqtam,

I'm pleased to inform you that your manuscript has been deemed suitable for publication in PLOS ONE. Congratulations! Your manuscript is now being handed over to our production team.

Kind regards,

on behalf of

Pro Fadwa Alhalaiqa

Academic Editor

PLOS ONE